# Effects of Lure Type on Chase-Related Behaviour in Racing Greyhounds

**DOI:** 10.3390/ani10122262

**Published:** 2020-12-01

**Authors:** Melissa Starling, Bethany Wilson, Paul McGreevy

**Affiliations:** Sydney School of Veterinary Science, University of Sydney, Camperdown, NSW 2050, Australia; bethany.wilson@sydney.edu.au (B.W.); paul.mcgreevy@sydney.edu.au (P.M.)

**Keywords:** predatory behaviour, canine, motivation, reinforcement

## Abstract

**Simple Summary:**

Contemporary racing greyhounds are bred chiefly to chase mechanical lures. In Australia, greyhounds are not commonly allowed access to the lure at the end of a race. It is unknown if, over time, this creates frustration that eventually reduces greyhounds’ willingness to chase a lure. A previous study found that greyhounds at race-meets often engage in behaviours, at the end of races, that may indicate frustration when the lure is still moving and audible but not accessible. Straight trial tracks have a quiet lure that stops much quicker than lures on racetracks. One straight trial track in New South Wales never allows greyhounds to access the lure while another track usually allows greyhounds to access the lure. This provides the opportunity to study behaviour in greyhounds before and after a chase where lures are accessible (straight track), not accessible and not detectable (straight track), and detectable but not accessible (racetrack). The results reveal that, compared with greyhounds at racetracks, those on straight tracks fixated more on the lure itself or where the lure was last seen. This suggests that the lure retains more salience if it is not associated with frustrating outcomes, such as protracted cues but no opportunity to chase.

**Abstract:**

The willingness of racing greyhounds in Australia to chase a mechanical lure on racetracks will affect the longevity of its racing career. Racing greyhounds that fail to chase may be retired from racing at an early age and their fate becomes uncertain and may in some cases be euthanasia. At the end of races, greyhounds are diverted into a catching pen while the lure continues on. Racing greyhounds may also run on straight tracks for training purposes, where the lure comes to a stop either within the catching pen or just outside it, rather than continuing on. The purpose of the current study was to determine if these different track conditions and lure features affected greyhound behaviour before and after chasing the lure. Video cameras were used to record the behaviour of greyhounds immediately before chasing a lure either on one of two straight trial tracks (*n* = 89 greyhounds) or during race-meets on oval racetracks (*n* = 537), as well as at the end of the chase in the catching pen. The results were analysed with logistic regression mixed models and coefficients expressed as odds ratios. It was predicted there would be a higher frequency of behaviours indicating frustration in the catching pen at tracks where no chase objects were accessible. This pattern was present, but not significant. It was also predicted there would be a higher frequency of behaviours that may indicate high anticipation before chasing at tracks where chase objects were accessible in the catching pen. This pattern was not realised. Behaviours prior to chasing varied between track types and days, suggesting these behaviours are unlikely to be good indicators of anticipation or motivation to chase. This study shows that greyhounds behave differently in the catching pen depending on the track and lure features.

## 1. Introduction

Greyhounds (*Canis familiaris*) are sighthounds and have been selectively bred for many generations to chase fast-moving visual stimuli. While they were originally bred to chase and capture prey such as hares, deer, rabbits, foxes and wolves [1], they also have a long history in the sport of coursing, which can be described as a competition between sighthound and live prey [1]. Coursing with live game still occurs in Éire, but has been outlawed in many other places, and has been widely replaced with racetracks and mechanical lures.

Despite many generations of selective breeding for chasing mechanical lures on a racetrack, some modern-day racing greyhounds do not reliably chase a mechanical lure. Failure to chase attracts penalties (endorsements) on the racetrack in Australia, in the form of temporary bars on racing, for offending individuals [2]. As a result, failure to chase contributes to greyhounds being retired from racing while still physically sound. This is an example of so-called behavioural wastage [3]. The proportion of greyhounds with endorsements for failure to chase in NSW is low, at fewer than 1% of starters (GRNSW, unpublished data), but it is likely that many dogs that do not chase reliably are retired from racing before they are given an official endorsement for failure to chase on the racetrack. 

The reasons for greyhounds failing to chase are not well understood, nor has the magnitude of this problem within the Australian greyhound racing industry been established. Indeed, aspects of canine chase responses are not well understood, in spite of their critical role in greyhound racing. For example, it is unknown to what extent chase behaviours are learned or innate, or how they develop in young dogs, at what age, how dogs may interact with chase objects when they have captured them, what reinforcers maintain or strengthen these behaviours, and what factors affect motivation to chase. This knowledge gap has relevance beyond greyhound racing, as chasing is a critical aspect of other working roles that dogs fill, such as herding livestock. Furthermore, there is potential for chasing behaviour to cause catastrophic problems such as sheep worrying. 

There is currently no published literature on chase motivation. Motivation in dogs for any resource or activity has been only sparsely considered in the literature. Motivation may be considered as processes of an organism that directs action towards the satisfaction of needs [4,5], and refers to the incentive to perform specific, goal-oriented behaviours. Motivation cannot be measured directly but must be inferred from an animal’s behaviour [6]. Greyhounds may chase lures because of their internal motivation to perform the behaviours involved and they derive direct reinforcement from the performance of those behaviours themselves (appetitive behaviours). Alternatively, greyhounds may be externally motivated to chase by the possibility of interacting with the lure (consummative behaviour). 

Rather than attempting to identify and track the minority of greyhounds that do not chase, an investigation into why greyhounds do chase allows for a larger pool of dogs to be drawn upon for data collection, and provides a starting point for identifying factors that contribute towards failure to chase. One potential factor in the maintenance of chase behaviours is whether greyhounds are sufficiently reinforced for chasing lures to continue doing so indefinitely. If greyhounds are reinforced by accessing objects they have been chasing, then preventing access may negatively affect interest in chasing lures as well as promote frustration. 

A previous study examined racing greyhound behaviour before and after races [7]. The study looked for behavioural signs of heightened arousal before races that may be related to performance or positive anticipation, and behavioural signs of frustration after races that may indicate a lack of reinforcement. Race-meets in Australia are held on oval tracks and greyhounds never get access to the lure at the end of a race. Instead they are funnelled into a sand-trap catching pen while the lure continues around the track. As such, they offer a model of how greyhounds behave in the absence of consummation.

The existence, in the Sydney region, of two operating straight tracks available for training purposes only (Appin and Redhead) offered an opportunity to collect additional data on greyhound behaviour prior to chasing and in catching pens under two conditions. The Appin track never allows greyhounds to access the lure, while the Redhead track usually allows greyhounds to access the lure in the catching pen. The current study was designed to quantify the behavioural differences in racing greyhounds immediately prior to and at the end of a chase according to whether or not they could access the lure. 

## 2. Materials and Methods

The University of Sydney Animal Ethics Committee approved the current study (Approval number: 2016/1015). The owners/handlers of the greyhounds provided informed consent for the collection of infrared images. 

### 2.1. Locations

The study was conducted at three greyhound racetracks in NSW over a period of 6 months, and two straight trial tracks in NSW over a period of 2 months. The racetracks were Richmond and Wentworth Park in the Sydney metropolitan area in June and July 2017 respectively, and Gosford on the NSW Central Coast, approximately 80 km north of Sydney, in October and November 2017. Data were collected from 3 race meets at Richmond, with 11 races per meet, 2 race meets at Wentworth Park with 10 races per meet, and 3 race meets at Gosford with 8, 10 and 11 races, respectively. Race number for each race meet was recorded so that it could be used as an indicator of how long dogs had spent at the race meet before infrared images were taken immediately prior to their race. The straight tracks were Appin, located approximately 70 km south-west of Sydney CBD, and Redhead, located approximately 150 km north of Sydney CBD.

The Appin and Redhead straight tracks operate for the purpose of training and conditioning racing greyhounds. These straight tracks are surfaced with grass, set on a gentle incline, and use a drag lure attached to a wire at ground level. Greyhounds are released from boxes or slipped (released by hand) and pursue the lure into a sand-trap catching pen at the end of the track. Both tracks were attended for the duration of their daily operation on two separate days in November and December in 2017. Dogs trialling at the straight tracks may be racing or in training, and are always muzzled. They start from boxes at two different distances (see details below) at both tracks, or they are slipped on the track near the start boxes. A description of the straight tracks is shown below.

#### 2.1.1. Appin

Start box distances: 274 m and 366 m. 

Lure: A teddy bear plush toy fixed to a metal sled, which is dragged up the centre of the track on a wire. The lure travels smoothly and does not bounce or leave the ground. 

Catching pen: Firm sand with a trapdoor on the back wall of the pen through which the lure exits the pen. The trapdoor closes behind the lure, so dogs cannot pass through.

Procedure: There is no formal stir-up on trial tracks. Trainers may get their dogs out of the trailer or vehicle several minutes before they run, and walk them on leash a short distance from the track, or they may unload dogs only when it is their turn to run. Dogs may be slipped on the track or started from boxes. Dogs usually run in singles, but up to 4 dogs can start at once. As the greyhounds approach the catching pen, the lure speed is increased to draw the lure away from the greyhound(s) chasing. This is intended to prompt the dogs to slow down. The lure comes to a stop behind a trapdoor at the back of the catching pen. A large gate descends vertically across the track at the start of the catching pen to contain dogs in the catching pen. Dogs are then caught by stewards or volunteers and placed in kennels alongside the catching pen until they are collected by their handlers. The handlers will water them and hose them down if desired, and then load them back into the car or trailer.

#### 2.1.2. Redhead

Start box distances: 250 m and 300 m.

Lure: A large polystyrene ball approximately 25 cm in diameter with a tail of synthetic fur attached. The lure is attached to a wire approximately 1 m from the left edge of the track. It bounces erratically as it is dragged up the track and hits small bumps in the ground. 

Catching pen: Soft sand-trap with a curtain approximately 1 m wide hung on the left-hand side of the pen halfway between the start of the sand-trap and the rear wall of the catching pen. The lure is brought to a halt behind the curtain.

Procedure: There is no formal stir-up on trial tracks. Trainers may get their dogs out of the trailer or vehicle several minutes before they run, and walk them on leash a short distance from the track, or they may unload dogs only when it is their turn to run. Dogs are either released from boxes or slipped on the track near the boxes. Track stewards report that the lure speed is increased as dogs approach the catching pen to encourage them to slow down, and the lure driver aims to stop the lure immediately behind the curtain in the catching pen. Dogs are prone to driving hard into the catching pen rather than slowing down. A gate can be pulled down vertically across the track to contain dogs in the catching pen, but this is not always performed. Dogs can travel around or through the curtain to access the lure. Stewards or volunteers catch dogs and transfer them to kennels on the left side of the catching pen where they are collected by their handlers. Again, the handlers will water them and hose them down if desired, and the dogs are then loaded back into the car/trailer.

#### 2.1.3. Racetracks

The data collected from Appin and Redhead tracks were pooled with behavioural data collected previously during race-meets at three different racetracks, Richmond, Wentworth Park, and Gosford [7]. All of these tracks adopt the same procedure for racing consisting of a stir-up where dogs about to race are allowed to watch the lure travel two laps of the track while they are in the stir-up pen beside the track. Greyhounds are walked to the start boxes once stir-up is over. The dogs are funnelled into the catching pen at the end of the race. Nominated catchers then catch the greyhounds and leash them. Richmond track has teaser toys attached to bungee cords in the catching pen. These cords are stretched across the track as the greyhounds approach the catching pen and then the toys released so that they bounce into the catching pen ahead of the greyhounds. The greyhounds therefore have the opportunity to interact with the teasers in the catching pen upon finishing the race. No lure, toy or teaser was available in the catching pens at Wentworth Park or Gosford. 

### 2.2. Subjects

A total of 525 greyhounds were recorded over the 8 race-meets at 3 racetracks. A total of 89 dogs were recorded at the straight tracks—43 at the Appin track and 46 at the Redhead track. Both male and female greyhounds were represented at all tracks, and dogs were aged 1–6 years old. Dogs varied in experience at racetracks, with their number of starts ranging 0–177. Experience of dogs at straight tracks was unknown as some had not attended race-meets. Dogs were excluded from data collection if they had previously been recorded by the current team of investigators at a prior meeting regardless of the track, so there were no repeat measures on dogs.

### 2.3. Behaviour Recording

The behaviour of the dogs was recorded using one GoPro Hero3 Black Edition action camera (GoPro, Inc. San Mateo, CA, USA) mounted onto the fence of the catching pen, and one hand-held Sony HD Handycam HDR-PJ760 video camera (Sony Corporation, Sony City, Minato, Tokyo, Japan). The videos were analysed in Windows Media Player 11 (Microsoft, Redmond, WA, USA) or QuickTime Player (Version 10.5 (1015.2.1), Apple Inc.,Cupertino, California, CA, USA) at 0.5 × speed. Sony HD Handycam HDR-PJ760 video cameras were used to film greyhounds before they were either slipped on the track or loaded into start boxes, starting when the greyhound was walked on-leash to the side of the track and finishing immediately after the dog was released to chase the lure. The ethograms developed for a related study on race day behaviour in greyhounds [7] were used again in this study. Five behaviours were added to the ethogram to record and characterise greyhounds interacting with the lure at the Redhead track—finish early, removed forcefully, paws, bite lure, and bite dog (Table 1). Behaviours that had recorded counts that were rarely higher than 2 were reduced to presence/absence variables, as shown in Table 1. Teasers were present in the catching pen at one racetrack (Richmond) and were used during races. Teasers are a toy comparable in size to a lure. They are usually made by wrapping foam around a length of PVC pipe and then securing a cover of synthetic fur to the outside. The purpose of a teaser is to encourage chasing by tossing it, dragging it along the ground, or wiggling it in the hand, and some dogs are given access to a teaser to bite and tug on it after chasing.

### 2.4. Statistical Analysis

All data were analysed by logistic mixed models using the glm and glmer functions from the lme4 library and the MASS package for ordinal regression in R (version 3.06, MA, USA; R Foundation for Statistical Computing, Vienna, Austria). The package ggplot2 was used to create graphs. Two models were trialled for each behaviour. The Full fitting factors were Sex of dog (Female/Male/Unknown); Time of day (Morning or Twilight/Evening); Distance of Race; Type of track (Straight or Race Track) with Track ID nested within Type, and Date nested within Track ID. Also included was whether an object is in the caching pen, and Track ID was nested within this also. The Reduced model removed Date nested within individual track. The reduced model was the more parsimonious as assessed by lower AIC and a non-significant deviance difference test in all cases. Standard residuals, and cooks distance were examined for influential outliers. The object in catching pen factor allows analysis of the effect of the availability of an object to focus on in the catching pen at the end of a chase and includes teaser (Richmond only), lure (Redhead only) and none (all other tracks). Wentworth Park and Gosford racetracks were not pooled for this analysis, as a previous study showed track differences between these tracks [7]. Wentworth racetrack was used as the reference track. Number of starts, distance travelled to reach the track, and age of dogs was only available for dogs at racetracks, but these factors were found to have no significant effect on behaviour at racetracks in a previous study [7].

Some behaviours in the ethogram in Table 1 did not occur with sufficient frequency to analyse formally. These included holding teaser, finish early, removed forcefully, paws, bite lure, and bite dog.

Teaser-related behaviours were uncommon compared to lure-related behaviours, and could only occur at Richmond racetrack, making analysis of these behaviours independent of lure-related behaviours challenging. It was decided to pool these behaviours along with holding teaser with fixation of the lure or lure gate so as not to lose all potential effects of the teaser on frustration-related behaviours through excluding them from analysis.

## 3. Results

Video data were collected from 89 dogs running at straight tracks (43 from Appin trial track and 46 from Redhead trial track), and 448 dogs running in races at race-meets (167 from Richmond racetrack, 153 from Wentworth Park racetrack, and 128 from Gosford racetrack).

Binary logistic regression models revealed that some behaviours were more likely to be observed at some tracks than others. Results are expressed as odds ratios (OR), which is a measure of the strength of the association between a condition and an outcome. The OR represents the odds that an outcome will occur, given the presence of a particular condition (e.g., the track, time of day, or the date) compared to the odds of that outcome occurring in the absence of that condition. An OR higher than 1 means an increased likelihood of the outcome occurring in the presence of the condition in question, whereas an OR lower than 1 means a decreased likelihood of the outcome occurring in the presence of the condition. Confidence intervals (CI) are also presented, which indicate the degree of uncertainty in the OR. A wide range in the CI indicates increased uncertainty. The tables in this section show OR and CI.

### 3.1. Catching Pen Behaviours

Results of analysis for the two most common catching pen behaviours—fixate on lure and jostling—are shown in Table 2. Fixation on the lure was significantly more likely at the straight tracks (OR = 8.34, z = 2.611, *p* = 0.009) than at racetracks (reference OR = 1), as shown in Figure 1. Additionally, the behaviour was observed much less commonly in the catching pen at Gosford than at the reference track of Wentworth Park (OR = 0.26, z = −2.512, *p* = 0.012). Jostling was also significantly less common at Gosford than at the reference track of Wentworth Park (OR= 0.29, z = −2.682, *p* = 0.007), but there was no significant differences between straight tracks and racetracks for this behaviour (z = 0.402, *p* = 0.688).

### 3.2. Stir-Up Behaviours

Three behaviours were observed both in the stir-up at racetracks and prior to box-loading or slipping at straight tracks—Handler-assisted rise, lunging, and jumping. The results of analysis of these behaviours is shown in Table 3. Handler assisted rise was less common at tracks with a chase object present in the catching pen (OR = 0.04, z = −3.796, *p* < 0.001), as shown in Figure 2, but the interaction term was also significant (OR = 6.79, z = 2.015, *p* = 0.044), suggesting that this reduction in the behaviour was smaller at Richmond with its teaser, than at Redhead with its lure. The difference between straight track and racetracks was not significant (z = 1.169, *p* = 0.243).

The difference between lunging at straight tracks and racetracks (z = −1.510, *p* = 0.131) and tracks with chase objects in the catching pen and those without (z = −1.810, *p* = 0.070), both fell short of statistical significance. Lunging was markedly less common at Gosford than at the reference Wentworth Park track, (OR = 0.21, z = −3.784, *p* < 0.001), the difference between these two tracks in the reference class for both comparisons potentially obscuring any differences. In addition, the odds of lunging were reduced (OR = 0.99, C.I. = 0.99–1.00) for every additional metre of race length.

The difference between jumping at straight tracks and racetracks (z = −0.817 *p* = 0.414) and tracks with objects in the catching pen and those without (z = −0606, *p* = 0.545), did not approach statistical significance. Once more, the behaviour was substantially less common at Gosford than at Wentworth Park.

## 4. Discussion

Contrasting the behaviour of dogs at straight tracks prior to trialling and in the catching pen with behaviour of dogs at race-meets in the stir-up and the catching pen identified some patterns that suggest stimulation that is important to greyhounds in chasing activities. These findings begin to characterise greyhounds’ interest in the lure.

### 4.1. Fixation on Chase Object

Greyhounds fixated on the lure or where it was last visible much more frequently at Redhead and Appin compared to the three racetracks. Racetrack lures are very loud. The noise mostly comes from the lure carriage traversing the rail at high speed, and this alone can be heard inside buildings at racetracks and even off the property. Racetrack lures also usually carry squawkers, which make quacking and/or chirping sounds loud enough to be heard over the lure carriage traversing the rail. Racetrack lures are not fitted with brakes and continue to move after the dogs have reached the catching pen. The visibility of the lure may also be limited, both at the start of races when dogs in starting boxes can only see out of a small window near the bottom of the box, and during the race where dogs at the front of the pack likely obscure view of the lure for dogs behind them. Thus, sound of the lure, rather than visual stimuli associated with it, may be the most salient stimulus in this environment. This has never been formally investigated, but sound is believed by a majority of racing greyhound industry participants in Australia to be more important than visual stimuli [3]. In contrast, the lure at straight tracks is quiet and stops moving quickly. For a greyhound in the catching pen on a straight track, there are no further signals pertaining to where the lure may be. It is either visible or neither audible nor visible. So, the situation on straight tracks compared to racetracks may more closely align to the ethology of canine predatory behaviour patterns, in that a pursuit ends either with capturing the prey, losing all sign of the prey, or with the prey close but inaccessible such as when it has taken refuge (e.g., underground) or travelled out of reach. Racetracks may represent a different end-of-chase for greyhounds, with a physical barrier preventing further pursuit as the chase object draws away.

There was a difference between frequency of fixation at the different racetracks, with this being less frequent at Gosford. Fixating on a teaser was pooled with fixating on the lure or lure gate for the current dataset, but this did not increase the percentage of dogs fixating at Richmond, where both teasers and the lure gate were present. This suggests that the tendency to fixate is most strongly influenced by proximity to the chase object and is not influenced by providing an alternative moving object. Richmond racetrack had a markedly lower frequency of all putative frustration behaviours in the catching pen compared to Gosford and Wentworth Park racetracks [7], and this may be associated with the teasers in the catching pen. That said, it would appear that the presence of the teasers may reduce putative frustration behaviours, but on the whole, a teaser is not a substitute for the lure, and most greyhounds will quickly abandon it. The difference between Gosford and Wentworth Park is more difficult to pinpoint. It is possible that there are features of the catching pens at racetracks that influence how willing greyhounds are to enter them, such as the level of lighting, the shape of the catching pen, or its proximity to other structures. If the greyhounds do not readily enter the catching pen, they remain on the track where they are closer to the gate where the lure disappeared, meaning they may be more likely to fixate on the lure gate if they were less likely to enter the catching pen in the first place.

Access to the lure may be granted on racetracks during training and at Redhead trial track, but was never granted at Appin trial track. In spite of this, fixating on the lure gate was not significantly different between Appin and Redhead, suggesting that this behaviour is not maintained or modulated by direct access to the lure. This aligns with chasing as a means to obtain food, where the behaviour often does not result in success. Hunting success rate among wild canids varies greatly but can be as low as 5.5% in dingoes [8] and less than 6% in Coyotes [9] while a range of 10–50% is reported in Grey Wolves [10]. A success rate of 66% has been reported when dingoes were observed chasing sheep, but dingoes studied to date have often been witnessed breaking-off a chase before the sheep is killed, suggesting that the dingoes in question were not sufficiently motivated to pursue a kill [8]. This is noteworthy, as it would appear dingoes’ motivation to chase may differ from their motivation to kill and subsequently feed.

Hunting success rate is poorly documented in domestic dogs. Domestic dogs assist human hunters most often by flushing prey animals, chasing them, and holding them at bay until human hunters catch up [11], meaning that the percentage of domestic dog hunting activities that result in prey capture may depend largely on the types of hunting they are bred for and engage in. Evidence from hare-coursing competitions in Éire suggests that greyhounds capture approximately 13% of hares in that context [12].

### 4.2. Jostling

Jostling in the catching pen was less frequent at Gosford than at Wentworth Park. An apparent pattern of increased frequency of this behaviour at tracks where there is no access to a chase object in the catching pen did not reach statistical significance. Jostling has not been validated as a behaviour associated with frustration or heightened arousal but, if it does occur primarily in association with these emotional states, it is expected that it would be most common at Wentworth Park and Gosford racetracks and Appin trial track and least common at Redhead trial track and Richmond racetrack where greyhounds could access the lure or a teaser in the catching pen. This expected pattern did not emerge in the current results, with jostling remaining low at Gosford. Jostling was infrequently recorded at the Redhead track on the first day of recording but more commonly recorded on the second day, which prevented it from being resolved in the statistical model. There was also considerable variation in jostling at Wentworth Park between observation days, and some variation at Gosford as well. Further data would be required to verify any influence of a track effect on this behaviour.

Jostling may also be affected by the number of dogs entering the catching pen. Where multiple dogs are exercised together on straight tracks, there are usually only two dogs involved; in contrast to the six to eight dogs involved in races. Jostling may also be influenced by the behaviour of other dogs in the catching pen. It is often accompanied by vocalising, which may encourage other dogs in the catching pen to jostle too. Another factor that may affect jostling is how familiar the dogs are with one another. Dogs that are familiar with one another may be less inhibited when together, and therefore more likely to jostle. However, familiarity is far more common on straight tracks (where dogs living at the same location are run together) than at oval racetracks, and there was no clear difference in jostling frequency between straight tracks and racetracks. Finally, jostling may also fluctuate in frequency in response to features of track design or prior experiences with specific tracks. For example, if jostling occurs as a response to frustration or anticipated frustration when losing sight of the lure or approaching the catching pen, then how early the catching pen becomes visible may signal coming frustration and thus prompt jostling with more time for dogs to engage in it. The shape of the catching pen varies between tracks and narrower or shorter catching pens may promote more jostling. There may also be an invisible effect of learning. If dogs have had experiences being jostled more often on a specific track, they may anticipate it and start it themselves as a result. It’s possible repeated races with many of the same dogs with similar track-specific experiences could create a culture of increased jostling at specific tracks.

### 4.3. Jumping, Lunging and Handler-Assisted Rise

Jumping and lunging were recorded before chases at all tracks. Jumping was less likely to occur at Gosford than at Wentworth Park, but there were no significant differences between tracks where chase objects were available in the catching pen and tracks where they were not. Lunging was less likely to occur at Redhead and Gosford tracks than at Wentworth Park. Lunging was also less likely to occur with longer chase distances. These behaviours were not associated with mean eye temperature at race-meets [7], so it is difficult to attribute them to arousal, as had been predicted when the data collection commenced. Instead of reflecting arousal, these behaviours may be directly related to trainers/handlers and the distance greyhounds are running. Of course, it is unlikely greyhounds know how long they will run before being released, but greyhounds that do not engage in energetically costly behaviours immediately prior to races may be better suited to running longer distances. As such, the negative relationship between the incidence of these behaviours and race distance may be due to dogs that are relatively inactive prior to races being better suited to longer races. An alternative explanation is that greyhounds prepared for longer races may be encouraged by trainers, however inadvertently, to display calmer behaviours prior to racing.

Handler-assisted rise was significantly less likely to occur at tracks with chase objects accessible in the catching pen. This is an intriguing pattern that runs counter to predictions of behaviours indicating high arousal before a chase potentially being indicative of anticipation and increased motivation to chase. As handler behaviour is a critical component of this, it may be that handlers are using this to try to increase arousal and anticipation in their dogs and are consciously or sub-consciously using it more at tracks without chase object accessibility. An alternative explanation is the handlers are assisting dogs to stand on their hind legs in response to the dog’s behaviour, but with different patterns emerging for jumping and lunging, it is unclear what that behaviour might be.

There is also the question of the role of age and experience on behaviours displayed in stir-up. A previous study found no effect of age or number of starts on behaviour indicative of heightened arousal in stir-up at racetracks, although there was a small, negative, but non-significant effect of the number of days since the dog last raced [7]. Race distance and track were the only significant factors influencing behaviours indicative of arousal in that study, which is consistent with the expanded dataset including straight tracks presented here.

These two behaviours occurred at different likelihoods at different tracks, and there was considerable variability in their occurrence. This may be indicative of confounding factors that have not been identified, such as handler influences or learning history with specific tracks. These behaviours may also be influenced by other transient factors such as how long it was since the dog has last trained, their current fitness, and even recent interactions with other dogs or humans that may have influenced their mood.

Finally, given that these behaviours may not stem from heightened arousal as originally thought, it is possible that they have differing functions in this context. Lunging is directed towards the lure whereas jumping is vertical motion without forward motion, suggesting the former could be goal-directed while the latter is not. However, the greyhounds are on leash at the time, so forward motion will almost certainly result in pressure on the collar whereas vertical motion alone probably will not, so differences could equally be attributable to sensitivity to collar pressure. It is beyond the scope of this study to identify the possible contributing factors to the differences between these behaviours observed.

### 4.4. Limitations to the Current Research

Data collection for this project was dependent on cooperation from greyhound owners and trainers and could not be allowed to impinge on normal operation of racetracks or trial tracks. Due to industry political and organisational factors beyond the researchers’ control, experimental research on the topic of chase motivation in greyhounds proved untenable. As such, data collection was limited to procedures that were fast and non-invasive and the data itself was limited to observational in nature.

Behavioural indicators of affective or motivational states in animals should be validated, preferably with physiological measures and/or many individuals in a battery of standardised tests covering a range of contexts. A comprehensive and validated ethogram for domestic dogs does not currently exist, so the behaviours singled out in this study should be considered likely candidates for further study rather than indicators of frustration, anticipation, motivation, or affective state.

Collecting data at racetracks and trial tracks only provided a snapshot of greyhound behaviour at various tracks. This behaviour may change as a greyhound becomes more experienced with racing and different tracks. For example, it was anecdotally suggested by stewards at Redhead trial track that greyhounds may learn after visiting several times to accelerate into the catching pen rather than decelerate as the lure-driving policy is supposed to encourage. This study did not collect data on acceleration of greyhounds or of the same greyhounds over several visits or at different tracks, so it was not possible to measure either changes in greyhound speed during chasing that may indicate an effect of the lure, or differences in greyhound behaviours across tracks that may reveal a lure or track effect, or changes in greyhound chase behaviour with experience.

### 4.5. Further Research

This study reveals that the location and stimuli associated with the lure at the end of a chase influences racing greyhound behaviour. The next step is to determine if these differences are meaningful to the greyhound’s experience of and value in racing—namely, whether the ability to fixate on a chase object is associated with performance and durability. Both performance and durability may hinge on whether the ongoing sound of the lure drawing away from greyhounds in the catching pen at the end of a chase heightens frustration in the dogs. Heightened frustration may lead to redirected aggression on conspecifics or human handlers, or to diminishing interest in chasing lures.

Including proxy measures of arousal, such as infrared thermography, before use of straight trial tracks as well as in greyhounds on racetracks that are often allowed to complete a trial in training by interacting with the lure may reveal whether the type of lure or likelihood of access to it affects anticipation. Fitting greyhounds with accelerometers before chasing would provide measurements of acceleration and deceleration on the track, which may provide evidence for how track type, catching pen features, or lure features influence chase behaviour in greyhounds.

Some greyhound owners and trainers using straight trial tracks anecdotally claim these tracks were a good tool for early lure-chasing training, and for giving racing greyhounds an enjoyable chase experience that may improve their performance at race-meets. Following individual dogs through their training and racing will likely provide valuable insights into how lure design and access may influence performance and reduce behavioural wastage.

Finally, chase behaviour is fundamental to working and companion dogs but, while in some working roles it is critical to success, it can also cause significant problems in companion dogs that chase inappropriate objects such as livestock or wildlife, other pets, cars, and humans either running or on bicycles, scooters or motorcycles. Inappropriate chasing can cause injury or death to the dog, animals they are chasing, and, in extreme circumstances, humans trying to avoid the dog. Despite the broadscale and sometimes acute problems that can stem from chasing behaviour in dogs, it has been subjected to very little scientific scrutiny. It is not known how chasing develops, how targets for chasing are learned or if they are innate, and the best methods for modifying chase behaviours. Dogs and humans alike would benefit from an ethological study into the development, manifestation and modification of chase behaviour in domestic dogs.

## 5. Conclusions

Comparing greyhound behaviour in stir-up and the catching pen at straight tracks versus racetracks and where chase objects are accessible or inaccessible in the catching pen revealed that greyhound behaviour differed depending on the type of track and whether an object was accessible in the catching pen. This difference was most clearly seen in a higher likelihood of fixating on the lure or where the lure was last visible in the catching pen at the end of the chase at straight tracks compared to racetracks. This result is likely to be influenced by cessation of sound and movement from the lure at straight tracks upon entering the catching pen, whereas at racetracks, the lure continues to move and emit a loud and characteristic sound after the dogs have entered the catching pen.

Jostling between dogs finishing together in the catching pen is considered a possible indicator of frustration in the catching pen, but this behaviour appears to be subject to high variability and possible chronological effects, meaning that more data on this behaviour may need to be collected to establish the conditions under which it is most likely to occur. Differences in track features may influence greyhound behaviour, especially in the catching pen.

Greyhound behaviours prior to the chase did not show clear patterns between tracks, and it is likely these behaviours are influenced most by the handler.

## Figures and Tables

**Figure 1 animals-10-02262-f001:**
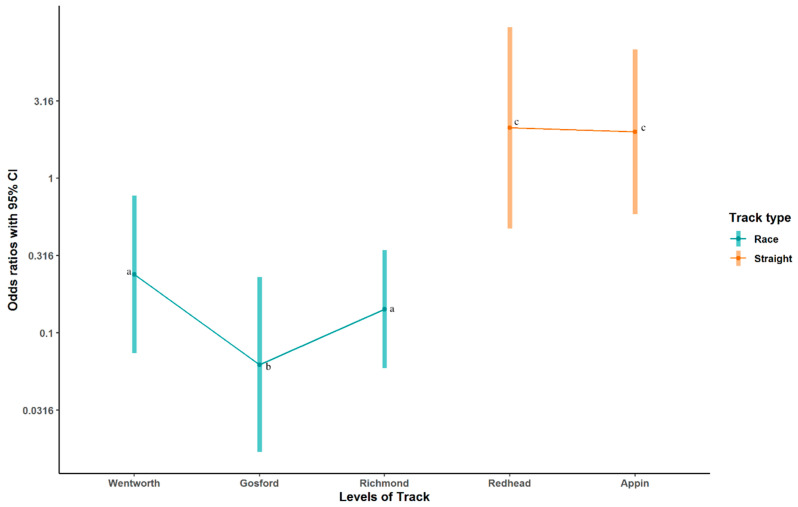
Odds ratios fixation on the lure or lure gate at straight tracks compared to racetracks. 95% confidence intervals are shown as error bars. This behaviour was significantly more likely at straight tracks than racetracks. Letters (a, b, c) denote results that are significantly different from each other.

**Figure 2 animals-10-02262-f002:**
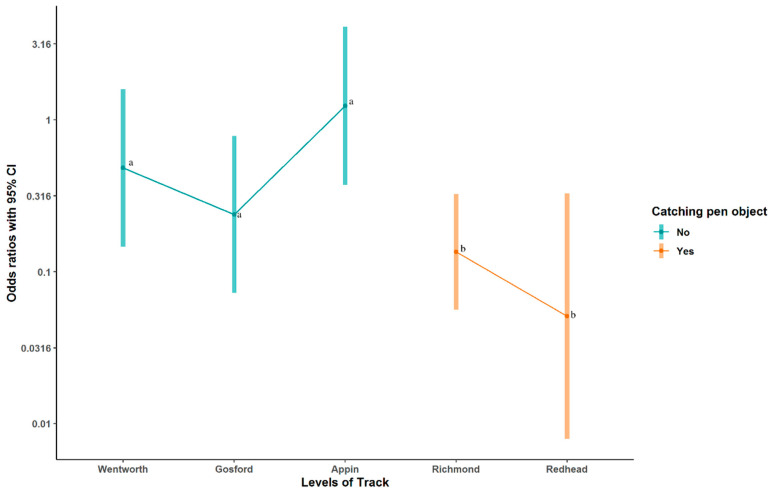
Odds ratios of Handler-assisted rise in tracks where a chase object was accessible in the catching pen and where a chase object was not accessible. 95% confidence intervals are shown as error bars. Letters (a, b) denote results that are significantly different from each other.

**Table 1 animals-10-02262-t001:** Ethogram of greyhound behaviours in catching pens on straight tracks. Behaviours additional to those recorded in Starling et al. (2020) are shown in italics.

Behaviour	Description	Record Type
Grabbing the teaser	Teeth contact the teaser but are released before trainers contact the dog.	Presence/absence
Changing directions	An approximate 180° change in direction while in motion.	Count
Jostling	Dog’s muzzle or shoulder makes physical contact with another dog with sufficient force to affect the trajectory of the receiving dog.	Count
Fixated on lure or lure gate	Dog orientating body position and interest towards the lure gate, or the lure if it is present.	Presence/absence
Holding teaser	Teaser is grabbed and not released by the greyhound.	Presence/absence
Finish early	Greyhounds come to a halt before reaching the lure gate or the lure.	Presence/absence
Removed forcefully	Greyhound does not release the lure until physically lifted off it by a steward, and/or is oriented towards the lure or lure gate as the steward moves them to the kennels beside the catching pen.	Presence/absence
Paws	Greyhound digs at the lure gate or grips or paws at the lure with one or two front paws.	Presence/absence
Bite lure	Greyhound’s mouth opens and closes while muzzle is in contact with the lure (Redhead only).	Count
Bite dog	Greyhound’s mouth opens and closes while muzzle is in contact with another dog (Redhead only).	Count

**Table 2 animals-10-02262-t002:** Model output for fixating on the lure or lure gate and jostling in the catching pen on straight tracks and racetracks compared to Wentworth Park as the reference track (racetrack). Statistically significant results are in bold and shaded. Results are statistically significant at *p* < 0.05 when 1.00 doesn’t fall between the 2.5% and 97% confidence interval.

Behaviour	Fixate on Lure			Jostling		
Odds-ratio	OR	2.50%	97.00%	OR	2.50%	97.00%
Catching pen object present- Lure i.e., Redhead (ref absent)	1.06	0.32	3.72	E	E	E
Track Type Straight (ref racetrack)	8.34	1.72	42.38 *	1.46	0.23	9.54
Time of Day Morning or Twilight (ref evening)	1.15	0.36	3.39	0.92	0.23	3.16
Distance (per meter)	1.00	0.99	1.00	1.00	0.99	1.00
Sex Unknown (reference female)	2.00	0.10	15.68	3.33	0.16	27.21
Sex Male (refence female)	0.65	0.36	1.18	1.12	0.59	2.18
Gosford (reference Wentworth)	0.26	0.08	0.71 *	0.29	0.11	0.70 *
Additional effect for Teaser Catching pen object i.e., Richmond	0.56	0.13	2.37	E	E	E

* denotes statistical significance.

**Table 3 animals-10-02262-t003:** Model output of behaviours occurring in the stir-up at straight tracks and racetracks. Statistically significant results are in bold and shaded. Results are statistically significant at *p* < 0.05 when 1.00 falls outside the 2.5% and 97% confidence interval and marked with an asterisk.

Behaviour	HA Rise			Lunging			Jumping		
Odds-ratio	OR	2.50%	97.00%	OR	2.50%	97.00%	OR	2.50%	97.00%
Catching pen Redhead (ref. absent)	0.04	0.01	0.18 *	0.13	0.01	0.85	0.62	0.12	2.71
Track Straight (ref. racetrack)	2.56	0.52	12.54	0.32	0.07	1.38	3.11	0.24	82.35
Time of Day Morning or Twilight (ref. evening)	1.84	0.58	5.82	0.74	0.28	1.80	0.64	0.03	5.39
Distance (per meter)	1.00	0.99	1.00	0.99	0.99	1.00 *	1.00	0.99	1.01
Sex Unknown (ref. female)	3.15	0.15	26.18	1.22	0.06	9.40	E	E	E
Sex Male (ref. female)	0.75	0.43	1.32	0.95	0.57	1.61	0.90	0.40	2.06
Gosford (ref. Wentworth)	0.49	0.21	1.11	0.21	0.09	0.46 *	0.23	0.05	0.83 *
Teaser Richmond	6.79	1.18	56.24 *	3.08	0.42	63.63	0.49	0.06	3.71

* denotes statistical significance.

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
