# Peer review of "Effects of Lure Type on Chase-Related Behaviour in Racing Greyhounds"

_animals, 2020, doi:10.3390/ani10122262_

Round 1

Reviewer 1 Report

The authors present important research regarding chase-related behaviour in racing greyhounds. Manuscript has been improved with this revision. Aside from minor typographical errors, the writing and content are appropriate.

1.Revise the table format of Table 3, such as frame lines and text-center.

2.I think it's better to put the significance marker on your figure.

Author Response

The authors present important research regarding chase-related behaviour in racing greyhounds. Manuscript has been improved with this revision. Aside from minor typographical errors, the writing and content are appropriate.

1.Revise the table format of Table 3, such as frame lines and text-center.

done

2.I think it's better to put the significance marker on your figure.

done

Reviewer 2 Report

You have sufficiently addressed the comments, thank you 

Author Response

You have sufficiently addressed the comments, thank you

Thanks.

This manuscript is a resubmission of an earlier submission. The following is a list of the peer review reports and author responses from that submission.

Round 1

Reviewer 1 Report

The authors aimed to explore the effects of track conditions and lure features on greyhound behavior before and after chasing the lure. The academic background of the manuscript is not comprehensive, that is, the author should give more information about greyhounds and related studies instead of redundant description of secular backgrounds and results in Introduction. The conclusion and discussion are relatively plain and weak. The authors should discuss the underlying reasons for difference between greyhound behavior at straight tracks and racetracks in depth rather than restate the results and limit the conclusion to the surface that behaviors DO have different patterns.

I am also baffled with the selection of reference track. Considering the teaser toy which only exists at the Richmond racetrack, it is inappropriate to select Richmond as the reference track. I strongly recommend you to merge the data from Gosford and Wentworth Park as reference track in your analysis. Finally, you should simplify your statement to clarify your idea efficiently in short sentences. See detailed comments below:

Simple Summary

  1. Line 15 Change “provided” to “provides”
  2. When the “greyhound” appears in the article first time, please provide its Latin name.
  3. Line 18 The preposition before “tracks” should be uniform throughout the manuscript.

Abstract

  1. Line 21-25 The background of greyhound should be shorten and simplified.
  2. Line 26-27 Rewrite the sentence “Straight trial tracks also exist……”
  3. The authors should clarify the meaning and significance of this study in a few sentences.

Keywords

  1. The keyword “reinforcement” seems to have a relatively low relation with the manuscript.

Introduction

  1. Line 44 Add “,” behind “[1]”
  2. Line 55 Add “is” behind “This”
  3. Line 61-64 Please rewrite the sentence “This knowledge gap applies beyond greyhound racing…”
  4. Line 67 Change “the” to “as”
  5. Line 82 Delete “are” in “are set on a gentle incline”
  6. Line 81- 86 Don't repeat the context about M&M

Materials and methods

  1. Line 94 Rewrite the sentence “The Appin and Redhead straight tracks operate on……”
  2. Line 98 Delete “may be” in “may be in training”
  3. Line 107 Delete “from”
  4. Line 109 Unify the format of citation in the main text.
  5. Line 110-111 Rewrite the sentence “with five additional behaviors being added to capture……”
  6. Line 128 Change “be started” to “start”
  7. Line 157 Rewrite the sentence “The lure is stopped, and the greyhounds are walked……”
  8. Line 166 Change “using linear mixed models” to “by linear mixed models”

Results

  1. Line 172 Change “at Redhead trial track” to “from Redhead trial track”
  2. Line 212 Change “0.988-0.98” to “0.988-0.998”
  3. The caption of Figure 1 and Figure 2 should be as concise as possible, deleting redundant description of results of analysis. Do not restate the results.
  4. There are some problems in the table format of Table 2 and Table 3. Please unify the format.
  5. Please state the statistically significant results of Distance in Results.

Discussion

The Discussion section is lack of logic which makes reading exceptionally difficult. Some instances listed in the Discussion have a weak connection with this study: e.g., line 259-262.

  1. Line 223 Change “it is” to “which is”
  2. Line 223-225 Shorten the description of the lure noise
  3. Please provide citations related to the effects of noise on racing behavior in this paragraph
  4. Line 239 “Fixating on a teaser was pooled with fixating on the lure or lure gate”, this treatment is inappropriate and groundless. The authors should not pool the behavior related to a teaser and the lure together.
  5. Line 272 Since the results of analysis do not fit with your expectation, further discussion on why this pattern comes out is needed.
  6. Line 284-286 Since familiarity could not explain why “there was no clear difference in jostling frequency between straight tracks and racetracks”, other possible reasons or factors should be provided.
  7. Line 289 Change “and” to “with”
  8. Further research needs to be shorten

Author Response

The authors thank the reviewers for their extensive comments. The reviewer comments are shown below with our responses in italics.

General comments about the Introduction and lack of clarity in the project aims have been addressed by adding the following to the end of the Introduction:

One potential factor in the maintenance of chase behaviours is whether greyhounds are sufficiently reinforced for chasing lures to continue doing so indefinitely. If greyhounds are reinforced by accessing objects they have been chasing, then preventing access may negatively affect interest in chasing lures as well as promote frustration.

A previous study examined racing greyhound behaviour before and after races [7]. The study looked for behavioural signs of heightened arousal before races that may be related to performance or positive anticipation, and behavioural signs of frustration after races that may indicate a lack of reinforcement. Race-meets in Australia are held on oval tracks and greyhounds never get access to the lure at the end of a race. Instead they are funnelled into a sand-trap catching pen while the lure continues around the track. As such, they offer a model of how greyhounds behave in the absence of consummation.

Some further references to behaviours prior to chasing have been added in the following paragraph to foreshadow the stir-up behaviour results.

Following comments from both Reviewers, the analysis has been redone using Gosford racetrack as the reference track and including a term for the availability of the chase object after the race – teaser (in catching pen, Richmond only), lure (Redhead only) and none (Gosford, Wentworth Park, and Appin). The following has been added to the Methods to reflect these changes:

Two models were trialled for each behaviour. The Full fitting factors were Sex of dog (Female/Male/Unknown); Time of day (Morning or Twilight/Evening); Distance of Race; Type of track (Straight or Race Track) with Track ID nested within Type, and Date nested within Track ID. Also included was whether an object is in the caching pen, and Track ID was nested within this also.  The Reduced model removed Date nested within individual track. The reduced model was the more parsimonious as assessed by lower AIC and a non-significant deviance difference test in all cases.  Standard residuals, and cooks distance were examined for influential outliers. The object in catching pen factor allows analysis of the effect of the availability of an object to focus on in the catching pen at the end of a chase and includes teaser (Richmond only), lure (Redhead only) and none (all other tracks). Wentworth Park and Gosford racetracks were not pooled for this analysis, as a previous study showed track differences between these tracks [7]. Wentworth racetrack was used as the reference track.

Reviewer 1

The authors aimed to explore the effects of track conditions and lure features on greyhound behavior before and after chasing the lure. The academic background of the manuscript is not comprehensive, that is, the author should give more information about greyhounds and related studies instead of redundant description of secular backgrounds and results in Introduction. The conclusion and discussion are relatively plain and weak. The authors should discuss the underlying reasons for difference between greyhound behavior at straight tracks and racetracks in depth rather than restate the results and limit the conclusion to the surface that behaviors DO have different patterns.

I am also baffled with the selection of reference track. Considering the teaser toy which only exists at the Richmond racetrack, it is inappropriate to select Richmond as the reference track. I strongly recommend you to merge the data from Gosford and Wentworth Park as reference track in your analysis. Finally, you should simplify your statement to clarify your idea efficiently in short sentences. See detailed comments below:

Simple Summary

  1. Line 15 Change “provided” to “provides”

This has been corrected.

  1. When the “greyhound” appears in the article first time, please provide its Latin name.

This has been added to Line 43.

  1. Line 18 The preposition before “tracks” should be uniform throughout the manuscript.

This has been checked throughout.

Abstract

  1. Line 21-25 The background of greyhound should be shorten and simplified.

This now reads:

The willingness of racing greyhounds in Australia to chase a mechanical lure on racetracks will affect the longevity of its racing career.

  1. Line 26-27 Rewrite the sentence “Straight trial tracks also exist……”

This now reads:

Racing greyhounds may also run on straight tracks for training purposes, where the lure comes to a stop either within the catching pen or just outside it, rather than continuing on.

  1. The authors should clarify the meaning and significance of this study in a few sentences.

Some sentences have been changed in communication of results in the Abstract. It now reads:

It was predicted there would be a higher frequency of behaviours indicating frustration in the catching pen at tracks where no chase objects were accessible. This pattern was present, but not significant. It was also predicted there would be a higher frequency of behaviours that may indicate high anticipation before chasing at tracks where chase objects were accessible in the catching pen. This pattern was not realised.

Keywords

  1. The keyword “reinforcement” seems to have a relatively low relation with the manuscript.

Additional information regarding the possible role of reinforcement in maintaining chase-related behaviours and thus extending racing greyhound careers in the Introduction should better illuminate why this keyword is relevant to the study.

Introduction

  1. Line 44 Add “,” behind “[1]”

Done

  1. Line 55 Add “is” behind “This”

Done

  1. Line 61-64 Please rewrite the sentence “This knowledge gap applies beyond greyhound racing…”

This now reads:

This knowledge gap has relevance beyond greyhound racing, as chasing is a critical aspect of other working roles that dogs fill, such as herding livestock. Furthermore,  there is potential for chasing behaviour to cause catastrophic problems such as sheep worrying.

  1. Line 67 Change “the” to “as”

Done

  1. Line 82 Delete “are” in “are set on a gentle incline”

Done

  1. Line 81- 86 Don't repeat the context about M&M

The following has been moved to the second paragraph of the Materials and Methods section:

These straight tracks are surfaced with grass, set on a gentle incline, and use a drag lure attached to a wire at ground level. Greyhounds are released from boxes or slipped (released by hand) and pursue the lure into a sand-trap catching pen at the end of the track.

Materials and methods

  1. Line 94 Rewrite the sentence “The Appin and Redhead straight tracks operate on……”

This sentence now reads:

The Appin and Redhead straight tracks operate for the purpose of training and conditioning racing greyhounds.

  1. Line 98 Delete “may be” in “may be in training”

Done

  1. Line 107 Delete “from”

Done

  1. Line 109 Unify the format of citation in the main text.

Done

  1. Line 110-111 Rewrite the sentence “with five additional behaviors being added to capture……”

This has been split into two sentences with the second reading:

Five behaviours were added to the ethogram to record and characterise greyhounds interacting with the lure at the Redhead track (shown in Table 1).

  1. Line 128 Change “be started” to “start”

Done

  1. Line 157 Rewrite the sentence “The lure is stopped, and the greyhounds are walked……”

This now reads:

Greyhounds are walked to the start boxes once stir-up is over. The dogs are funnelled into the catching pen at the end of the race.

  1. Line 166 Change “using linear mixed models” to “by linear mixed models”

Done

Results

  1. Line 172 Change “at Redhead trial track” to “from Redhead trial track”

Done

  1. Line 212 Change “0.988-0.98” to “0.988-0.998”

Done

  1. The caption of Figure 1 and Figure 2 should be as concise as possible, deleting redundant description of results of analysis. Do not restate the results.

Reiteration of results has been removed from figure captions.

  1. There are some problems in the table format of Table 2 and Table 3. Please unify the format.

Unclear of nature of format issues.

  1. Please state the statistically significant results of Distance in Results.

The following sentence has been added to section 3.2. of the Results:

In addition, the odds of lunging were reduced (OR=0.99, C.I. =0.99-1.00) for every additional metre of race length.

Discussion

The Discussion section is lack of logic which makes reading exceptionally difficult. Some instances listed in the Discussion have a weak connection with this study: e.g., line 259-262.

  1. Line 223 Change “it is” to “which is”

This change would be more obfuscating, so the sentence has been changed to:

The noise mostly comes from the lure carriage traversing the rail at high speed

  1. Line 223-225 Shorten the description of the lure noise

This now reads:

Racetrack lures also usually carry squawkers, which make quacking and/or chirping sounds loud enough to be heard over the lure carriage traversing the rail.

  1. Please provide citations related to the effects of noise on racing behavior in this paragraph

There are no citations because it has never been researched. The following has been added to this paragraph to clarify:

The visibility of the lure may also be limited, both at the start of races when dogs in starting boxes can only see out of a small window near the bottom of the box, and during the race where dogs at the front of the pack likely obscure view of the lure for dogs behind them. Thus, sound of the lure, rather than visual stimuli associated with it, may be the most salient stimulus in this environment. This has never been formally investigated, but sound is believed by a majority of racing greyhound industry participants in Australia to be more important than visual stimuli [3].

  1. Line 239 “Fixating on a teaser was pooled with fixating on the lure or lure gate”, this treatment is inappropriate and groundless. The authors should not pool the behavior related to a teaser and the lure together.

This decision was made because there were too many confounds and too few dogs interested in teasers to examine behaviour towards teasers independently, but ignoring it all together seems inappropriate given that it may influence track effects considerably. This has now been explained in the Materials and Methods section with the following:

Teaser-related behaviours were uncommon compared to lure-related behaviours, and could only occur at Richmond racetrack, making analysis of these behaviours independent of lure-related behaviours challenging. It was decided to pool these behaviours with fixation of the lure or lure gate so as not to lose all potential effects of the teaser on frustration-related behaviours through excluding them from analysis.

  1. Line 272 Since the results of analysis do not fit with your expectation, further discussion on why this pattern comes out is needed.

Several potential confounds are already referred to. The following has been added:

Finally, jostling may also fluctuate in frequency in response to features of track design or prior experiences with specific tracks. For example, if jostling occurs as a response to frustration or anticipated frustration when losing sight of the lure or approaching the catching pen, then how early the catching pen becomes visible may signal coming frustration and thus prompt jostling with more time for dogs to engage in it. The shape of the catching pen varies between tracks and narrower or shorter catching pens may promote more jostling. There may also be an invisible effect of learning. If dogs have had experiences being jostled more often on a specific track, they may anticipate it and start it themselves as a result. It’s possible repeated races with many of the same dogs with similar track-specific experiences could create a culture of increased jostling at specific tracks. 

  1. Line 284-286 Since familiarity could not explain why “there was no clear difference in jostling frequency between straight tracks and racetracks”, other possible reasons or factors should be provided.

It is beyond the scope of the project to determine whether familiarity affected behaviour at different tracks, as this study only gathered information on greyhounds once. It would be necessary to gather information on individual greyhounds multiple times at multiple tracks to determine if there was a stable track effect on their behaviour. The number of dogs racing together at straight tracks vs racetracks is considered the most likely explanation for the lack of resolution in the data, and this has already been raised.

  1. Line 289 Change “and” to “with”

Done

  1. Further research needs to be shorten

The authors are reluctant to shorten this section as it addresses current knowledge gaps. Given this is the first study to examine chase-related behaviours in a working dog application, the authors feel it is important to outline each step that needs to be taken both within the racing greyhound context to adequately answer the question of chase motivation and why greyhounds fail to chase lures, but also understand that chase-related behaviours have broader implications that deserve further study.

Reviewer 2 Report

This study presented an examination of the behaviour of Greyhounds at different track environments which provides novel and thought-proking information pertaining to chase motivation in these working dogs. The study contributes to the literature on Greyhound racing and the potential impact of the design of tracks and lures. 

However, the results are presented in such a way that makes it difficult to draw conclusions about the influence of lure access in line with the study objective. 

This study aimed to quantify the behavioural differences in greyhounds at the end of a chase according to whether or not they could access the lure and whether or not the lure was still detectable, but not accessible (Line 90). However, the paper also described behaviours pre-racing (not just at the end of a chase) in the stir up before a race. There is no mention of the stir up process on the straight tracks in the methods and it is not clear how different this phase was compared to the oval racetracks. In addition, the statistical comparisons were made for all racetracks compared to Richmond (where a teaser is accessible) rather than according to whether a lure/teaser was accessible (redhead, richmond) or not (appin, wentworth and Gosford). There is insufficient investigation or discussion as to whether behavioural differences were influenced by other differences between tracks or not, for example the different types of lure (teddy bear at Appin versus Redhead), number of dogs run together or stir up processes.  

I would therefore like to recommend a different statistical model design be used, where the outcome of lure access is modelled controlling for track site, which will allow clearer interpretation of the influence of access to the lure on observed behaviours. Lure type would also warrant investigation as it may have an impact on chase motivation. 

Specific comments below. 

Line 55 – missing an “is”… “this an example”

Line 61 – you describe that canine chase responses are not well understood but there are published literature on chase behaviour and you include some in the discussion (8, 10, 11). This statement is therefore misleading.

Line 98 since straight tracks are commonly used for training, could the age or experience of the dog have an impact on the motivation to chase? This has not been sufficiently discussed in this paper.

Line 152 – there is not enough detail in the methods of data collection for racetracks as compared to data collection from the straight tracks. The number of dogs was expressed in the methods for straight tracks so it would be consistent to report here too (though it is later reported in the results). It is not clear if the data from different races includes the same dogs – if this is a possibility, there might be some individual dog effect, which would need to be considered and where possible, adjusted for in the analysis.

Line 165 – there is not enough detail in the statistical analysis methods – which variables were included in analysis - were any confounding variables included or controlled for? In the results, you include time of day, distance and sex of the dog in your models. These data are not well described in the methods - were any other variables measured?

There are no P-values presented – I might assume you are using 95% probability but I would prefer to see P-values reported alongside OR. Indeed, for Table 2 – for the near significance finding for jostling at Appin compared to Richmond, you refer to the table but there are no P-values in the table so I cannot see how near this was to significance.

Figures – it would be helpful to have a visual marker to show the significant findings

Line 204 missing “were” – three “were” behaviours observed

Results in general – there is no mention of the other behaviours recorded in Table 1 – were these included in the statistical models? If not, why not and if so, were they not significant?

Line 29 – when discussing the association between Jumping and lunging to arousal – is eye temperature is not a validated measure of arousal? Could other measures of arousal be investigated before this is ruled out? You tend to group jumping and lunging and discuss them both but findings suggest their likelihood was different at different tracks – what might be the reason for this? Why do might they be grouped under the same motivation?

Line 294 – you remark that greyhounds are unlikely to know how long they will run before released so this is not likely an appropriate point for discussion, unless you can explain your meaning. Perhaps a more appropriate explanation is that age or experience may play a role here?

Discussion – general. Please discuss the high variability (as indicated by the high CI) in your findings, particularly for stir-up behaviours (e.g. jumping at Appin). How might the high variability be explained? 

Author Response

The authors thank the reviewers for their extensive comments. The reviewer comments are shown below with our responses in italics.

General comments about the Introduction and lack of clarity in the project aims have been addressed by adding the following to the end of the Introduction:

One potential factor in the maintenance of chase behaviours is whether greyhounds are sufficiently reinforced for chasing lures to continue doing so indefinitely. If greyhounds are reinforced by accessing objects they have been chasing, then preventing access may negatively affect interest in chasing lures as well as promote frustration.

A previous study examined racing greyhound behaviour before and after races [7]. The study looked for behavioural signs of heightened arousal before races that may be related to performance or positive anticipation, and behavioural signs of frustration after races that may indicate a lack of reinforcement. Race-meets in Australia are held on oval tracks and greyhounds never get access to the lure at the end of a race. Instead they are funnelled into a sand-trap catching pen while the lure continues around the track. As such, they offer a model of how greyhounds behave in the absence of consummation.

Some further references to behaviours prior to chasing have been added in the following paragraph to foreshadow the stir-up behaviour results.

Following comments from both Reviewers, the analysis has been redone using Gosford racetrack as the reference track and including a term for the availability of the chase object after the race – teaser (in catching pen, Richmond only), lure (Redhead only) and none (Gosford, Wentworth Park, and Appin). The following has been added to the Methods to reflect these changes:

Two models were trialled for each behaviour. The Full fitting factors were Sex of dog (Female/Male/Unknown); Time of day (Morning or Twilight/Evening); Distance of Race; Type of track (Straight or Race Track) with Track ID nested within Type, and Date nested within Track ID. Also included was whether an object is in the caching pen, and Track ID was nested within this also.  The Reduced model removed Date nested within individual track. The reduced model was the more parsimonious as assessed by lower AIC and a non-significant deviance difference test in all cases.  Standard residuals, and cooks distance were examined for influential outliers. The object in catching pen factor allows analysis of the effect of the availability of an object to focus on in the catching pen at the end of a chase and includes teaser (Richmond only), lure (Redhead only) and none (all other tracks). Wentworth Park and Gosford racetracks were not pooled for this analysis, as a previous study showed track differences between these tracks [7]. Wentworth racetrack was used as the reference track.

Reviewer 2

This study presented an examination of the behaviour of Greyhounds at different track environments which provides novel and thought-proking information pertaining to chase motivation in these working dogs. The study contributes to the literature on Greyhound racing and the potential impact of the design of tracks and lures. 

However, the results are presented in such a way that makes it difficult to draw conclusions about the influence of lure access in line with the study objective. 

This study aimed to quantify the behavioural differences in greyhounds at the end of a chase according to whether or not they could access the lure and whether or not the lure was still detectable, but not accessible (Line 90). However, the paper also described behaviours pre-racing (not just at the end of a chase) in the stir up before a race. There is no mention of the stir up process on the straight tracks in the methods and it is not clear how different this phase was compared to the oval racetracks. In addition, the statistical comparisons were made for all racetracks compared to Richmond (where a teaser is accessible) rather than according to whether a lure/teaser was accessible (redhead, richmond) or not (appin, wentworth and Gosford). There is insufficient investigation or discussion as to whether behavioural differences were influenced by other differences between tracks or not, for example the different types of lure (teddy bear at Appin versus Redhead), number of dogs run together or stir up processes.  

I would therefore like to recommend a different statistical model design be used, where the outcome of lure access is modelled controlling for track site, which will allow clearer interpretation of the influence of access to the lure on observed behaviours. Lure type would also warrant investigation as it may have an impact on chase motivation. 

Some additions regarding stir-up have been made in the Introduction, and the data have been re-analysed to change the reference track and include a factor of chase objects accessible in the catching pen (see general comments). The following sentences were added to the “Procedure” section of the straight track descriptions:

There is no formal stir-up on trial tracks. Trainers may get their dogs out of the trailer or vehicle several minutes before they run, and walk them on leash a short distance from the track, or they may unload dogs only when it is their turn to run.

Specific comments below. 

Line 55 – missing an “is”… “this an example”

Already addressed

Line 61 – you describe that canine chase responses are not well understood but there are published literature on chase behaviour and you include some in the discussion (8, 10, 11). This statement is therefore misleading.

This statement has been changed to reflect what exactly is not well understood about chase behaviours in canines. It now reads:

Indeed, aspects of canine chase responses are not well understood, in spite of their critical role in greyhound racing. For example, it us unknown to what extent chase behaviours are learned or innate, or how they develop in young dogs, at what age, how they may interact with chase objects when they have captured them, what reinforcers maintain or strengthen these behaviours, and what factors affect motivation to chase.

Line 98 since straight tracks are commonly used for training, could the age or experience of the dog have an impact on the motivation to chase? This has not been sufficiently discussed in this paper.

Greyhounds continue to train throughout their racing career, and some continue to attend straight tracks regularly or irregularly while some trainers use them mostly in early training. It was beyond the scope of this study to examine experience with specific tracks, whether they be straight tracks or racetracks, which is stated in the Limitations section. Earlier work on greyhounds at racetracks showed age and number of starts had no significant effect on the behaviours recorded in stir-up or eye temperature before races, and that dataset included dogs that had no prior starts as well as veterans of over 100 starts. We do not have sufficient data from this study to discuss motivation to chase, so this point is instead addressed in the methods section with the following:

All data were analysed by logistic mixed models using the glm and glmer functions from the lme4 library and the MASS package for ordinal regression in R (version 3.06, MA, USA; R Foundation for Statistical Computing, Vienna, Austria). The package ggplot2 was used to create graphs. Two models were trialled for each behaviour. The Full fitting factors were Sex of dog (Female/Male/Unknown); Time of day (Morning or Twilight/Evening); Distance of Race; Type of track (Straight or Race Track) with Track ID nested within Type, and Date nested within Track ID. Also included was whether an object is in the caching pen, and Track ID was nested within this also.  The Reduced model removed Date nested within individual track. The reduced model was the more parsimonious as assessed by lower AIC and a non-significant deviance difference test in all cases.  Standard residuals, and cooks distance were examined for influential outliers. The object in catching pen factor allows analysis of the effect of the availability of an object to focus on in the catching pen at the end of a chase and includes teaser (Richmond only), lure (Redhead only) and none (all other tracks). Wentworth Park and Gosford racetracks were not pooled for this analysis, as a previous study showed track differences between these tracks [7]. Wentworth racetrack was used as the reference track. Number of starts, distance travelled to reach the track, and age of dogs was only available for dogs at racetracks, but these factors were found to have no significant effect on behaviour at racetracks in a previous study [7].

Line 152 – there is not enough detail in the methods of data collection for racetracks as compared to data collection from the straight tracks. The number of dogs was expressed in the methods for straight tracks so it would be consistent to report here too (though it is later reported in the results). It is not clear if the data from different races includes the same dogs – if this is a possibility, there might be some individual dog effect, which would need to be considered and where possible, adjusted for in the analysis.

See the addition from the previous point. The following sections have also been added to the Methods:

2.1. Locations

The study was conducted at three greyhound racetracks in NSW over a period of 6 months, and two straight trial tracks in NSW over a period of 2 months. The racetracks were Richmond and Wentworth Park in the Sydney metropolitan area in June and July 2017 respectively, and Gosford on the NSW Central Coast, approximately 80 km north of Sydney, in October and November 2017. Data were collected from 3 race meets at Richmond, with 11 races per meet, 2 race meets at Wentworth Park with 10 races per meet, and 3 race meets at Gosford with 8, 10 and 11 races, respectively. Race number for each race meet was recorded so that it could be used as an indicator of how long dogs had spent at the race meet before infrared images were taken immediately prior to their race. The straight tracks were Appin, located approximately 70km south-west of Sydney CBD, and Redhead, located approximately 150km north of Sydney CBD.

The Appin and Redhead straight tracks operate for the purpose of training and conditioning racing greyhounds. These straight tracks are surfaced with grass, set on a gentle incline, and use a drag lure attached to a wire at ground level. Greyhounds are released from boxes or slipped (released by hand) and pursue the lure into a sand-trap catching pen at the end of the track.

And separately:

2.2. Subjects

A total of 525 greyhounds were recorded over the 8 race meets at 3 racetracks. A total of 89 dogs were recorded at the straight tracks – 43 at the Appin track and 46 at the Redhead track. Both male and female greyhounds were represented at all tracks, and dogs were aged 1–6 years old. Dogs varied in experience at racetracks, with their number of starts ranging 0–177. Experience of dogs at straight tracks was unknown as some had not attended race-meets. Dogs were excluded from data collection if they had previously been recorded by the current team of investigators at a prior meeting regardless of the track, so there were no repeat measures on dogs.

Line 165 – there is not enough detail in the statistical analysis methods – which variables were included in analysis - were any confounding variables included or controlled for? In the results, you include time of day, distance and sex of the dog in your models. These data are not well described in the methods - were any other variables measured?

See the addition from earlier comment, added at line 222.

There are no P-values presented – I might assume you are using 95% probability but I would prefer to see P-values reported alongside OR. Indeed, for Table 2 – for the near significance finding for jostling at Appin compared to Richmond, you refer to the table but there are no P-values in the table so I cannot see how near this was to significance.

It is doubtful we could easily fit another two columns into the tables, so this has been addressed by adding the following to table captions:

Results are statistically significant at p<0.05 when 1.00 doesn’t fall between the 2.5% and 97% confidence interval.

Figures – it would be helpful to have a visual marker to show the significant findings

Figures have been updated with a statement of significance in the captions.

Line 204 missing “were” – three “were” behaviours observed

Corrected, thank you.

Results in general – there is no mention of the other behaviours recorded in Table 1 – were these included in the statistical models? If not, why not and if so, were they not significant?

Some of the behaviours in Table 1 did not occur often enough for analysis. The following has been added to the Results section to explain:

Some behaviours in the ethogram in Table 1 did not occur with sufficient frequency to analyse formally. These included holding teaser, finish early, removed forcefully, paws, bite lure, and bite dog.

Line 29 – when discussing the association between Jumping and lunging to arousal – is eye temperature is not a validated measure of arousal? Could other measures of arousal be investigated before this is ruled out? You tend to group jumping and lunging and discuss them both but findings suggest their likelihood was different at different tracks – what might be the reason for this? Why do might they be grouped under the same motivation?

A previous study recently published included eye temperature at racetracks alone. It was found that eye temperature did not have a significant relationship with the pre-race behaviours thought to indicate heightened arousal, which is already addressed in the Discussion. The following has been added to the relevant section of the Discussion:

Finally, given that these behaviours may not stem from heightened arousal as originally thought, it is possible that they have differing functions in this context. Lunging is directed towards the lure whereas jumping is vertical motion without forward motion, suggesting the former could be goal-directed while the latter is not. However, the greyhounds are on leash at the time, so forward motion will almost certainly result in pressure on the collar whereas vertical motion alone probably will not, so differences could equally be attributable to sensitivity to collar pressure. It is beyond the scope of this study to identify the possible contributing factors to the differences between these behaviours observed.

Line 294 – you remark that greyhounds are unlikely to know how long they will run before released so this is not likely an appropriate point for discussion, unless you can explain your meaning. Perhaps a more appropriate explanation is that age or experience may play a role here?

The point has been clarified and the relationship between age, experience, and behaviours indicating arousal explained in more detail with the following addition to the relevant section of the Discussion:

As such, the negative relationship between the incidence of these behaviours and race distance may be due to dogs that are relatively inactive prior to races being better suited to longer races. An alternative explanation is that greyhounds prepared for longer races may be encouraged by trainers, however inadvertently, to display calmer behaviours prior to racing.

Handler-assisted rise was significantly less likely to occur at tracks with chase objects accessible in the catching pen. This is an intriguing pattern that runs counter to predictions of behaviours indicating high arousal before a chase potentially being indicative of anticipation and increased motivation to chase. As handler behaviour is a critical component of this, it may be that handlers are using this to try to increase arousal and anticipation in their dogs and are consciously or sub-consciously using it more at tracks without chase object accessibility. An alternative explanation is the handlers are assisting dogs to stand on their hind legs in response to the dog’s behaviour, but with different patterns emerging for jumping and lunging, it is unclear what that behaviour might be.

There is also the question of the role of age and experience on behaviours displayed in stir-up. A previous study found no effect of age or number of starts on behaviour indicative of heightened arousal in stir-up at racetracks, although there was a small, negative, but non-significant effect of the number of days since the dog last raced [7]. Race distance and track were the only significant factors influencing behaviours indicative of arousal in that study, which is consistent with the expanded dataset including straight tracks presented here.

Discussion – general. Please discuss the high variability (as indicated by the high CI) in your findings, particularly for stir-up behaviours (e.g. jumping at Appin). How might the high variability be explained? 

The following has been added to the Discussion:

These two behaviours occurred at different likelihoods at different tracks, and there was considerable variability in their occurrence. This may be indicative of confounding factors that have not been identified, such as handler influences or learning history with specific tracks. These behaviours may also be influenced by other transient factors such as how long it was since the dog has last trained, their current fitness, and even recent interactions with other dogs or humans that may have influenced their mood. 

Finally, given that these behaviours may not stem from heightened arousal as originally thought, it is possible that they have differing functions in this context. Lunging is directed towards the lure whereas jumping is vertical motion without forward motion, suggesting the former could be goal-directed while the latter is not. However, the greyhounds are on leash at the time, so forward motion will almost certainly result in pressure on the collar whereas vertical motion alone probably will not, so differences could equally be attributable to sensitivity to collar pressure. It is beyond the scope of this study to identify the possible contributing factors to the differences between these behaviours observed.